# 3D-GRAND: A Million-Scale Dataset for 3D-LLMs with Better Grounding and Less Hallucination

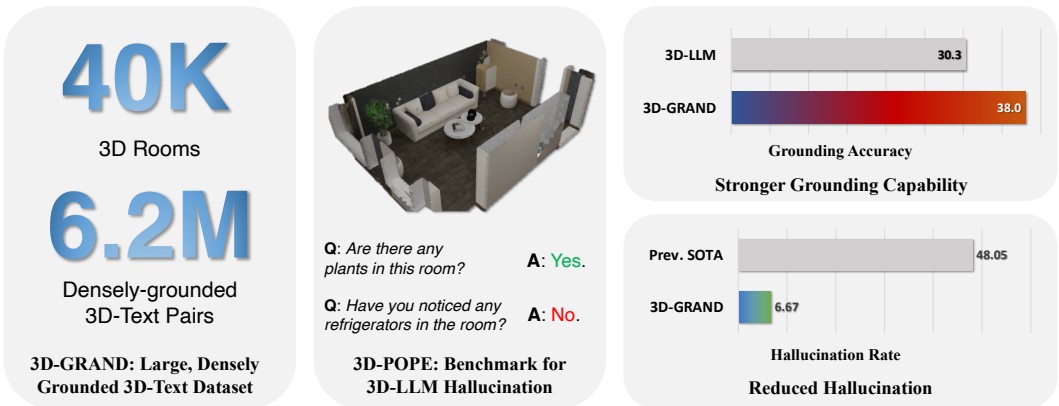

Figure 1: We introduce 3D-GRAND, a large-scale, densely grounded 3D-text dataset, and 3D-POPE, a 3D-LLM hallucination benchmark. Training on 3D-GRAND improves grounding accuracy and reduces hallucinations.

## Abstract

The integration of language and 3D perception is crucial for developing embodied agents and robots that comprehend and interact with the physical world. While large language models (LLMs) have demonstrated impressive language understanding and generation capabilities, their adaptation to 3D environments (3D-LLMs) remains in its early stages. A primary challenge is the absence of large-scale datasets that provide dense grounding between language and 3D scenes. In this paper, we introduce 3D-GRAND, a pioneering large-scale dataset comprising 40,087 household scenes paired with 6.2 million densely-grounded scene-language instructions. Our results show that instruction tuning with 3D-GRAND significantly enhances grounding capabilities and reduces hallucinations in 3D-LLMs. As part of our contributions, we propose a comprehensive benchmark 3D-POPE to systematically evaluate hallucination in 3D-LLMs, enabling fair comparisons among future models. Our experiments highlight a scaling effect between dataset size and 3D-LLM performance, emphasizing the critical role of large-scale 3D-text datasets in advancing embodied AI research. Notably, our results demonstrate early signals for effective sim-to-real transfer, indicating that models trained on large synthetic data can perform well on real-world 3D scans. Through 3D-GRAND and 3D-POPE, we aim to equip the embodied AI community with essential resources and insights, setting the stage for more reliable and better-grounded 3D-LLMs.

## 1 Introduction

Embodied Artificial Intelligence (EAI) represents a frontier in robotics and machine learning. In EAI, the integration of perception, language, and action within physical spaces is crucial for developing intelligent systems capable of meaningfully navigating and interacting with their environments. Central to this vision is the concept of *grounding* language in the physical world (Bisk et al., 2020;

Chandu et al., 2021). Grounding connects abstract linguistic constructs to concrete objects in three-dimensional space, thereby enabling robots and intelligent agents to effectively understand and manipulate their surroundings.

Recent advancements in Large Language Models (LLMs) have greatly benefited Embodied Artificial Intelligence (EAI). LLMs demonstrate exceptional capabilities in understanding language instructions (OpenAI, 2024b; Touvron et al., 2023), perceiving the environment (Liu et al., 2023; Li et al., 2023a; Alayrac et al., 2022; Zhu et al., 2023a; Yang et al., 2024a), and planning detailed actions (Brohan et al., 2023; Huang et al., 2023d). The primary inputs to LLMs, other than pure language, have been the combination of language and 2D images, categorizing these models as 2D-LLMs. The significant advancements in 2D-LLMs can be largely attributed to their training on extensive vision-language datasets. These datasets (Schuhmann et al., 2022; Zhu et al., 2023b), comprising billions of image and text pairs, have been instrumental in enhancing the models' understanding of visual content and its contextual relevance to textual information. These large datasets have provided the foundational data necessary for training models that excel at integrating visual perception with language processing. Despite some progress in equipping LLMs to understand 3D scenes (3D-LLMs) (Hong et al., 2023b; Huang et al., 2023a; Wang et al., 2023b; Huang et al., 2024; Zhu et al., 2023c; Chen et al., 2023; Qi et al., 2023), these models remain in their early stages due to the scarcity of 3D scene and text pairs. In this work, we introduce 3D-GRAND, a pioneering million-scale dataset designed for densely-grounded 3D Instruction Tuning.

Recently, SceneVerse (Jia et al., 2024) concurrently introduced a large-scale 3D vision-language dataset. However, a significant limitation of this dataset is the absence of object grounding in language, which is crucial for enhancing model usability in robotics tasks and reducing hallucination. Research on 2D-LLMs indicates that grounding language to 2D contexts notably mitigates hallucination in language models (You et al., 2023; Peng et al., 2023; Bai et al., 2023; Lai et al., 2023; Rasheed et al., 2023; Zhang et al., 2024), thereby enhancing the reliability and interpretability of generated responses. While 2D grounding has been extensively explored, extending these principles to 3D environments is still underdeveloped. This situation raises two critical questions: (1) *Is there any hallucination in 3D-LLMs and if so, how severe is it*? (2) *Can densely-grounded data mitigate hallucination for 3D-LLMs*? These questions underscore a critical need within the research community for the development of an evaluation benchmark specifically designed for 3D-LLMs and the construction of a large-scale, 3D-grounded dataset.

To quantify hallucination in 3D LLMs, this work introduces 3D-POPE (3D Polling-based Object Probing Evaluation). 3D-POPE provides a comprehensive and standardized protocol for evaluating hallucination that enables systematic assessment and facilitates fair comparisons across 3D-LLMs, enhancing our understanding of model capabilities in object hallucination. Specifically, we pose existence questions to 3D-LLMs and evaluate their responses, as illustrated in Fig 1.

To evaluate the role of densely-grounded dataset, we introduce a pioneering million-scale dataset, 3D-GRAND, for densely grounded 3D instruction tuning. 3D-GRAND includes 40,087 household scenes paired with 6.2 million scene-language instructions, featuring dense phrase-to-object grounding. We conduct rigorous human evaluations to ensure the dataset's quality. Our results trained with 3D-GRAND highlight the dataset's effectiveness in enhancing grounding and reducing hallucination for 3D-LLMs. We highlight the effectiveness of incorporating 3D-GRAND in Fig 1 and introduce each category of 3D-GRAND and provide examples in Fig 2.

To sum up, our contributions include:

- 3D-GRAND, the first **million-scale**, **densely-grounded** 3D-text dataset for grounded 3D Instruction Tuning. 3D-GRAND includes 40K household scenes paired with 6.2M densely-grounded scene-language instructions.

- 3D-POPE, a suite of benchmarks and metrics that systematically evaluate hallucination, enabling fair comparisons of future 3D-LLM models in terms of object hallucination.

- Quantitative research findings regarding hallucination, grounding, and scaling that provide guidance to future research: (1). training 3D-LLMs with 3D-GRAND significantly reduces hallucinations, particularly when the data is densely grounded; (2). densely grounded instruction tuning significantly enhances the grounding capabilities of 3D-LLMs; (3). scaling densely grounded data consistently improves grounding accuracy and reduces hallucination; and (4). models can successfully transfer from sim-to-real, providing an early signal for a low-cost and sustainable future of scaling synthetic 3D data to help on real tasks.

| Dataset | Which part is grounded? | Densely Grounded? | Language source | # 3D Scenes | # Language pairs |
|---|---|---|---|---|---|
| ReferIt3D (Achlioptas et al., 2020) | obj-refer | ✗ | Human,Template | 0.7K | 125K |
| ScanRefer (Chen et al., 2020) | obj-refer | ✗ | Human | 0.7K | 51K |
| Scan2Cap (Chen et al., 2021) | obj-refer | ✗ | Human | 0.7K | 51K |
| ScanEnts3D (Abdelreheem et al., 2024) | obj-refer | ✓ | Human | 0.7K | 84K |
| PhraseRefer (Yuan et al., 2022) | obj-refer | ✓ | Human | 0.7K | 170K |
| ScanQA (Azuma et al., 2022) | answer | ✗ | Human | 0.7K | 41K |
| SQA3D (Ma et al., 2023) | question | ✗ | Human | 0.65K | 33.4K |
| 3DVQA (Etesam et al., 2022) | ✗ | ✗ | Template | 0.7K | 500K |
| CLEVR3D (Yan et al., 2021) | ✗ | ✗ | Template | 8.7K | 171K |
| 3DMV-VQA (Hong et al., 2023a) | ✗ | ✗ | Template | 4.1K | 55K |
| EmbodiedScan (Wang et al., 2023a) | ✗ | ✗ | Template | 3.4K | 970K |
| 3DMIT (Li et al., 2024) | ✗ | ✗ | LLM | 0.7K | 75K |
| M3DBench (Li et al., 2023b) | obj-refer, question | ✗ | LLM | 0.7K | 327K |
| 3D-DenseOG (Huang et al., 2023c) | scene | ✓ | Human | 0.7K | 51K |
| 3D-LLM (Hong et al., 2023b) | obj-refer | ✗ | LLM | 0.9K | 200K |
| LL3DA (Chen et al., 2023) | question, answer | question | Template,LLM | 0.9K | 200K |
| Chat3D-v2 (Huang et al., 2023a) | scene | ✓ | Human,LLM | 0.7K | 0.7K |
| 3D-VisTA (Zhu et al., 2023c) | question | ✗ | Template,LLM | 3K | 278K |
| LEO (Huang et al., 2024) | question | ✗ | LLM | 3K | 579K |
| SceneVerse (Jia et al., 2024) | obj-refer | ✗ | Template,LLM | 62K | 2.5M |
| **3D-GRAND** | scene, obj-refer, question, answer | ✓ | Template,LLM | 40K | 6.2M |

Table 1: Comparison of 3D-GRAND with existing 3D scene datasets with language annotations. 3D-GRAND is the largest language-grounded dataset.

## 2 RELATED WORK

**Injecting 3D into LLMs.** Recent advancements in large language models (LLMs) have inspired research into extending their capabilities to 3D environments, leading to the development of 3D-LLMs (Chen et al., 2023; Qi et al., 2023; Yang et al., 2024a; Zhu et al., 2023c). Notable works in this field include 3D-LLM (Hong et al., 2023b), which integrates 3D point clouds and features into LLMs to enable tasks such as captioning, question answering, and navigation. LEO (Huang et al., 2024) excels as an embodied multi-modal generalist agent in perception, grounding, reasoning, planning, and action in 3D environments, highlighting the potential of 3D-LLMs in understanding and interacting with the physical world. The most relevant work to our model is Chat-3Dv2 (Wang et al., 2023b; Huang et al., 2023a), which grounds generated scene captions to objects in 3D scenes. However, Chat-3Dv2's dataset is limited to one type of 3D-text task (scene captioning) and only includes 705 captions from a subset of ScanNet scenes. In 3D-GRAND, we expand this concept by diversifying 3D-text tasks and increasing the dataset size to a million-scale. Our results demonstrate promising data scaling effects and sim-to-real transfer, paving the way for future large-scale training of 3D-LLMs.

**Object Hallucination of VLMs.** While 2D VLMs have achieved impressive performance, they are prone to hallucinating objects that do not exist in the provided images, a problem known as *object hallucination* (Dai et al., 2023; Rohrbach et al., 2018). Several methods have been suggested to mitigate the object hallucination issue, such as integrating an external object detector zhai2023halle, applying visually grounded visual instruction tuning you2023ferret,zhang2024groundhog or reinforcement learning sun2023aligning,gunjal2024detecting, performing iterative refinement zhou2023analyzing, and adapting the decoding strategies huang2023opera. To quantify and mitigate this issue, several benchmarks have been proposed. CHAIR (Caption Hallucination Assessment with Image Relevance) (Rohrbach et al., 2018) measures the frequency of hallucinated objects in image captions by comparing the objects mentioned to the ground truth annotations. POPE (Probing Object Hallucination Evaluation) (Li et al., 2023c) assesses a VLM's ability to identify the presence or absence of objects through yes/no probing questions. However, these studies primarily focus on 2D image-text datasets like COCO (Lin et al., 2014). In contrast, object hallucination in 3D-LLMs remains largely unexplored. Our work addresses this gap by introducing 3D-POPE, a comprehensive benchmark for evaluating object hallucination in 3D-LLMs. To the best of our knowledge, this is the first object hallucination benchmark for 3D-LLMs.

**Grounding Datasets for 3D-LLMs.** In the 2D domain, large-scale datasets with grounding information have been instrumental in advancing vision-language research. Notable examples include RefCOCO (Yu et al., 2016), which provides referring expressions for objects in COCO images (Lin et al., 2014). Additionally, 2D LLMs (Peng et al., 2023; Rasheed et al., 2023; Xu et al., 2023; Lai et al., 2023; You et al., 2023) have been trained with densely-grounded web-crawled image-text pairs. In the 3D domain, there is a growing interest in creating datasets that pair 3D scenes with textual annotations (Yuan et al., 2022; Abdelreheem et al., 2024; Huang et al., 2023c; Chen et al., 2021). ScanRefer (Chen et al., 2020) pioneered this effort by contributing a dataset of ScanNet (Dai et al., 2017) scenes with referring expressions. Table 1 introduces the efforts made by the community.

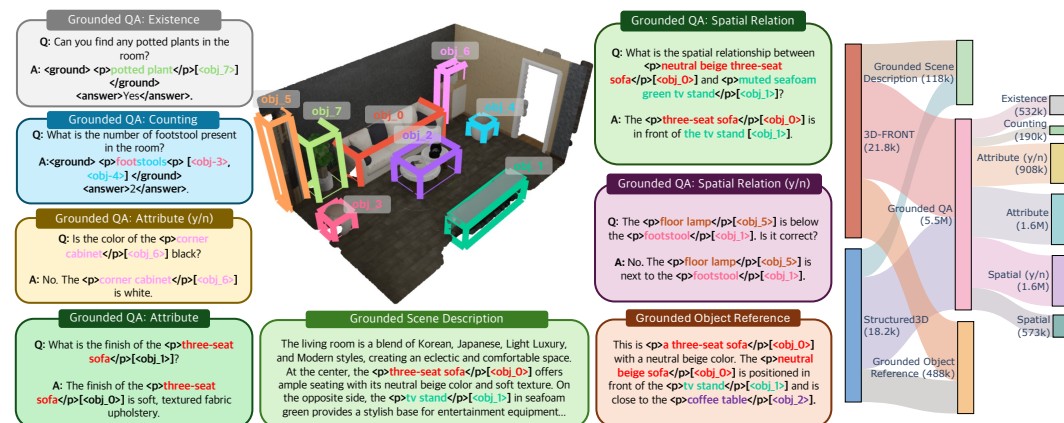

Figure 2: 3D-GRAND dataset and statistics. **(Left):** 3D-GRAND is a large-scale, densely-grounded 3D-text dataset with 8 different tasks. **(Right):** From 40K 3D scenes, 3D-GRAND annotates 6.2M 3D-text pairs.

However, these datasets have limited grounding annotations and often focus on a single task, such as referring expression comprehension or visual question answering. In contrast, our proposed dataset, 3D-GRAND, stands out by providing 6.2 million densely-grounded scene-language instructions across a diverse set of 3D-text tasks and 40,087 household scenes. This enables a wide range of grounding tasks and facilitates the development of more reliable and better-grounded 3D-LLMs.

Among recent datasets, 3D-GRAND is most similar to SceneVerse (Huang et al., 2023c). They are both million-scale grounding datasets for 3D-LLMs. However, there are a few key differences: (1) SceneVerse (Huang et al., 2023c) provides only sparse grounding, while 3D-GRAND is densely grounded. In 3D-GRAND, every noun phrase in the text—whether it's in captions, QAs, or object references—is explicitly grounded to a corresponding object in the 3D scene, whereas SceneVerse does not offer this level of grounding granularity. To elucidate this difference, we present Table 2 and 3 that compares SceneVerse and 3D-GRAND; (2) the language annotations of 3D-GRANDare more trustworthy and have higher quality. Hallucination is known as one of the most common mistakes of LLMs (Huang et al., 2023b; Li et al., 2023c; Rohrbach et al., 2018) In 3D-GRAND, we employ a hallucination filter to check and delete any annotations with hallucinated object IDs. This is not possible for SceneVerse since they have pure language output. 3D-GRAND is also quality-checked by humans to ensure the quality.

|  | Scene Caption | Object Reference | QA |
|---|---|---|---|
| **SceneVerse** | Paragraph-level set-to-set grounding | Session-level many-to-one grounding | No grounding |
| **3D-GRAND** | Noun-level one-to-one grounding | Noun-level one-to-one grounding | Noun-level one-to-one grounding |

Table 2: Comparison of grounding granularity in SceneVerse and 3D-GRAND.

|  | Grounding Granularity | Object Reference Data |
|---|---|---|
| **SceneVerse** | Session-level many-to-one grounding | This is a big cotton sofa. It is between the window and the wooden table. → *sofa-3* |
| **3D-GRAND** | Noun-level one-to-one grounding | This is a ***big cotton sofa*** [*sofa-3*] . **It** [*sofa-3*] is between the **window** [*window-0*] and **wooden table** [*table-4*] . |

Table 3: Example of grounding granularity. 3D-GRAND focuses on dense grounding.

Definitions of grounding granularity:

- **Paragraph-level set-to-set grounding**: Many sentences in a long paragraph, each containing several object nouns, are linked to a set of 3D objects without clear associations from specific sentences/noun phrases to objects.
- **Session-level many-to-one grounding**: Multiple sentences in one session, where each sentence can describe several objects (targets and landmarks), are associated with one 3D object.
- **Noun-level one-to-one grounding**: Each noun phrase in each sentence is explicitly matched with one 3D object.

## 3   3D-GRAND: THE 3D GROUND ANYTHING DATASET

In this section, we introduce 3D-GRAND, a large-scale, densely-grounded 3D-text dataset designed for grounded 3D instruction tuning. We describe the data collection process, dataset statistics, and the unique features that make 3D-GRAND a valuable resource for advancing research in 3D-LLMs.

**3D scene collection.** The majority of 3D-text research is currently based on ScanNet scenes collected from real camera scans, which are limited in scale. However, recent advancements have led to the development of numerous synthetic data generation pipelines (Mittal et al., 2023; Deitke et al., 2020; Ehsani et al., 2021; Deitke et al., 2022; Kolve et al., 2017; Puig et al., 2023; Szot et al., 2021; Manolis Savva* et al., 2019; Yang et al., 2024b; Höllein et al., 2023; Schult et al., 2023b; Juliani et al., 2020; Epic Games). Given the scalability of these synthetic data generation pipelines, we explore the potential of using synthetic 3D scenes to enhance 3D-text understanding.

Synthetic data offers significant advantages, such as lower costs and greater accessibility, making it an attractive alternative. If models trained on simulated 3D-text data can effectively transfer to real-world 3D scenes, the research community stands to benefit immensely.

To this end, we curate a diverse collection of 40,087 high-quality 3D indoor scenes from the 3D-FRONT (Fu et al., 2021) and Structured3D (Zheng et al., 2020) datasets. These datasets are chosen for their large quantities of synthetic indoor scenes with professionally designed layouts. The collection includes a variety of room types, such as living rooms, bedrooms, kitchens, office spaces, and conference rooms. We further process these 3D scenes to generate per-room 3D point clouds. Details on point cloud rendering and cleaning are provided in the Appendix.

**Densely-grounded text annotation.** The definition of *densely-grounded* text is that every noun phrase of object mentioned in the text should be associated with an 3D object in the 3D scene. This is illustrated in Figure 2. This is a difficult type of data to get annotations on. Early work such as ScanEnts3D (Abdelreheem et al., 2024) relies on hiring *professional* human annotators to obtain such annotations. The authors report that crowd-sourcing annotators (Amazon Mechanical Turk (AMT) (Crowston, 2012)) were not able to reliably complete this task and they had to hire professional annotators (error rate AMT: 16%, professional: <5%). Yet our human quality check shows that LLMs (GPT-4 (OpenAI, 2024b)) can achieve <8.2-5.6% densely-grounding error rate (see Appendix for detail). This finding is in accordance with recent studies (Ding et al., 2023; Tan et al., 2024) reporting LLMs can be human-level annotators. The accuracy of LLM-annotation provides one motivation for considering LLMs as densely grounding annotation tool.

The second, and perhaps more critical, motivation is the scalability of annotation. While we can potentially scale up 3D scenes using synthetic data generation pipelines, annotating these scenes with human effort is both costly and time-consuming, especially for complex tasks like densely grounding annotation. To put the cost of money and time in perspective, for the data we annotated in this paper, we estimate that obtaining the same annotations with human annotator would cost at least $539,000 and require 5.76 years (no eat, no sleep) worth of work from a professional annotator (earning minimum wage of $10.67 per hour). In contrast, using LLMs (GPT4 (OpenAI, 2024b)), we achieve the same results for $3,030 within 2 days, representing a 178x reduction in cost and a 1051x reduction in time. At the time of writing, the cost and time further decreases by 50% to $1,500 and 1 day, with the introduction of GPT-4o (OpenAI, 2024a).

As previously discussed, using humans to annotate 3D scenes can be an exhaustive process. Meanwhile, 2D-LLMs demonstrate remarkable capabilities in understanding visual inputs and generating language, making them well-suited for creating high-quality, grounded language annotations. However, due to the hallucination issues and data issues in 2D-LLMs, aggregating information across images, even those originating from the same scene, is not feasible yet.

In contrast, Large Language Models (LLMs) excel at understanding structural data and generating diverse and fluent language (OpenAI, 2024b). They have demonstrated capabilities in spatial reasoning (Bubeck et al., 2023), solving both elementary and sophisticated math problems (Wu et al., 2023; Imani et al., 2023). To address the limitations of 2D-LLMs when annotate 3D scenes, we leverage the strengths of LLMs. By integrating detailed, accurate information into a reliable scene graph, we provide LLMs with the necessary data to reason effectively and generate precise annotations.

Here are the key steps of applying our pipeline to obtain densely-grounded annotation for any synthetic 3D scene:

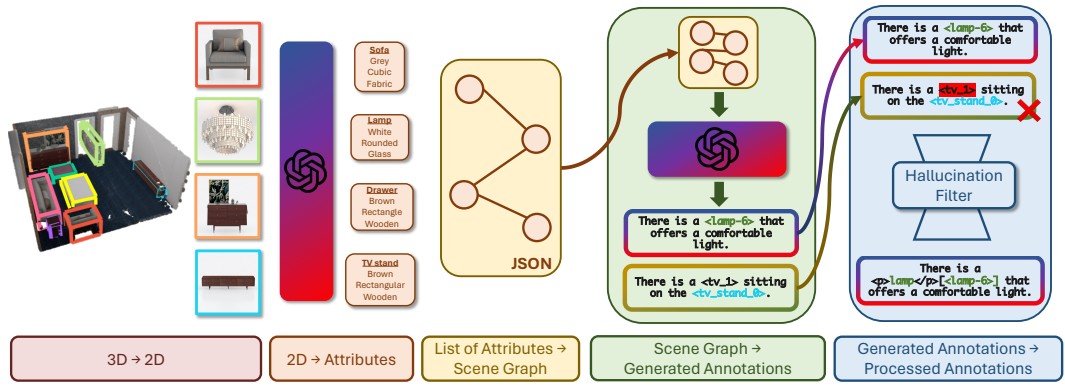

Figure 3: 3D-GRAND Data Curation Pipeline.

• **3D Model to 2D Image.** In the 3D-Front dataset, each object is sourced from 3D-Future (Fu et al., 2021), which provides a ground truth 2D image for each object. For the Structured3D dataset, individual images for each object are not available. Therefore, we utilize the set-of-mark prompting technique (Yang et al., 2023), where each object to be annotated is circled in red in the images.
• **2D Image to Attributes.** We use GPT-4V to generate detailed language annotations for each 2D object image, including attributes like name, color, finish, and texture. The naming is now open-vocabulary, contrary to being class-agnostic.
• **List of Attributes to Scene Graph.** We structure each individual objects' annotations into a JSON-based scene graph that captures the relationships and attributes of objects within the scene. Note that we obtain this scene graph from synthetic data which we can guarantee the correctness.
• **Scene Graph to Generated Annotations.** Based on the given scene graph, we will be able to produce 3D-Grounded Object Reference, 3D-Grounded Scene Description, and 3D-Grounded QA using GPT-4 (OpenAI, 2024b) with various prompts, which we will show in the appendix.
• **Generated Annotations to Processed Annotations.** After we acquire raw annotations, we will apply hallucination filters and template augmentation for the phrase tag to remove low-quality annotations and augment generated annotations.

With this pipeline, we generate a diverse range of 3D vision-language understanding tasks as shown in Figure 2. On a high level, these tasks can be categorized into:

• **3D-Grounded Object Reference**: Given a 3D scene and an object of interest, 3D-LLM is required to generate a description that uniquely identifies the target object. The description includes text and grounding information, not only for the target object but also for any landmark objects mentioned in the description. This task is conceptually similar to Visual Grounding, Scene-aware Object Captioning, and Dense Captioning in 2D vision-language research.
• **3D-Grounded Scene Description**: Given a 3D scene, the 3D-LLM generates a description that captures the salient aspects of the environment. The description includes both text and grounding information, linking the language to specific objects or regions in the scene.
• **3D-Grounded QA**: Given a 3D scene and a question about the environment, the 3D-LLM generates an answer that is grounded in the scene. Both the question and answer include text and grounding information, ensuring that the 3D-LLM's responses are contextually relevant and accurate.

**Human quality check.** As shown in Table 4, we conducted extensive human quality checks on 5,100 generated annotations, comparing the error rates of our dataset, 3D-GRAND, with those of previous datasets such as ScanEnts3D (Abdelreheem et al., 2024). The results demonstrate that large language models (LLMs), such as GPT-4, can achieve error rates in densely grounded annotations comparable to those of professional human annotators. This finding aligns with recent studies that suggest LLMs are starting to reach human-level annotation quality on certain tasks (Tan et al., 2024). See Appendix for a more detailed description of the human quality check process and results.

| Annotation Source | Error Rate |
|---|---|
| ScanEnts3D (AMT) | 16% |
| ScanEnts3D (Professional) | <5% |
| 3D-GRAND (LLM, GPT-4) | 5.6-8.2% |

Table 4: Error rates comparison between ScanEnts3D and 3D-GRAND annotations. (AMT = Amazon Mechanical Turk)

| Dataset | 3D-POPE | Model | Precision | Recall | F1 Score | Accuracy | Yes (%) |
|---|---|---|---|---|---|---|---|
| | | Random Baseline | 50.00 | 50.00 | 50.00 | 50.00 | 50.00 |
| | | 3D-LLM (Hong et al., 2023b) | 50.03 | 99.88 | 66.67 | 50.07 | 99.81 |
| | *Random* | 3D-VisTA (Zhu et al., 2023c) | 50.12 | 53.58 | 51.79 | 49.66 | 53.95 |
| | | LEO (Huang et al., 2024) | 51.95 | 77.65 | 62.25 | 52.91 | 74.73 |
| | | Ours zero-shot (Grounding) | **93.34** | 84.25 | 88.56 | 89.12 | 45.13 |
| | | Random Baseline | 50.00 | 50.00 | 50.00 | 50.00 | 50.00 |
| | | 3D-LLM (Hong et al., 2023b) | 49.97 | 99.88 | 66.61 | 49.94 | 99.94 |
| ScanNet200 Val | *Popular* | 3D-VisTA (Zhu et al., 2023c) | 47.40 | 51.88 | 49.54 | 49.49 | 52.30 |
| | | LEO (Huang et al., 2024) | 48.30 | 77.65 | 59.55 | 47.27 | 80.38 |
| | | Ours zero-shot (Grounding) | **73.05** | 84.28 | 78.26 | 76.59 | 57.69 |
| | | Random Baseline | 50.00 | 50.00 | 50.00 | 50.00 | 50.00 |
| | | 3D-LLM (Hong et al., 2023b) | 49.97 | 99.88 | 66.61 | 49.94 | 99.94 |
| | *Adversarial* | 3D-VisTA (Zhu et al., 2023c) | 48.28 | 54.39 | 51.15 | 51.14 | 52.99 |
| | | LEO (Huang et al., 2024) | 48.47 | 77.98 | 59.78 | 47.52 | 80.45 |
| | | Ours zero-shot (Grounding) | **69.86** | 84.21 | 76.37 | 73.95 | 60.26 |

Table 5: 3D-POPE benchmark results for evaluating hallucination in 3D language models. Random Baseline refers to a model randomly predicting "yes" or "no" with 50% chance, given the 1:1 positive/negative sample ratio in the dataset.

**Dataset highlights.** 3D-GRAND possesses several unique features that distinguish it from existing 3D-language datasets: (1). *Large-scale*: With 40,087 scenes and 6.2 million annotations, 3D-GRAND is the largest 3D-language dataset to date, providing ample data for training and evaluating 3D-LLMs. (2). *Dense grounding*: Unlike recent million-scale datasets like SceneVerse, which lack grounded language annotations, each language annotation in 3D-GRAND is densely grounded to specific objects or regions within the 3D scenes, facilitating fine-grained language understanding and generation. (3). *Diverse language tasks*: 3D-GRAND supports a broad array of grounded language tasks, including object reference , spatial reasoning, and scene understanding, making it a comprehensive benchmark for evaluating 3D-LLMs. (4). *High-quality annotations*: We utilize a hallucination filter to mitigate hallucination of the language annotations in 3D-GRAND. They are also human-evaluated to ensure the quality.

These unique features establish 3D-GRAND as a valuable resource for advancing research in 3D-LLMs and embodied AI. By providing a large-scale, densely-grounded 3D-text dataset, 3D-GRAND enables the development and evaluation of more capable and reliable 3D-LLMs that can effectively understand and interact with the physical world.

# 4 3D-POPE: A BENCHMARK FOR EVALUATING HALLUCINATION IN 3D-LLMS

To systematically evaluate the hallucination behavior of 3D-LLMs, we introduce the 3D Polling-based Object Probing Evaluation (3D-POPE) benchmark. 3D-POPE is designed to assess a model's ability to accurately identify the presence or absence of objects in a given 3D scene.

**Dataset.** To facilitate the 3D-POPE benchmark, we curate a dedicated dataset from the ScanNet (Dai et al., 2017) dataset, utilizing the semantic classes from ScanNet200 (Rozenberszki et al., 2022). Specifically, we use the ScanNet validation set as the foundation for evaluating 3D-LLMs on the 3D-POPE benchmark.

**Benchmark design.** 3D-POPE consists of a set of triples, each comprising a 3D scene, a posed question, and a binary answer ("Yes" or "No") indicating the presence or absence of an object (Fig. 1 middle). To ensure a balanced dataset, we maintain a 1:1 ratio of existent to nonexistent objects when constructing these triples. For the selection of negative samples (i.e., nonexistent objects), we employ three distinct sampling strategies:

- **Random Sampling**: Nonexistent objects are randomly selected from the set of objects not present in the 3D scene.

- **Popular Sampling**: We select the top-$k$ most frequent objects not present in the 3D scene, where $k$ equals the number of objects currently in the scene.

- **Adversarial Sampling**: For each positively identified object in the scene, we rank objects that are not present and have not been used as negative samples based on their frequency of co-occurrence with the positive object in the training dataset. The highest-ranking co-occurring object is then selected as the adversarial sample. This approach differs from the original POPE (Li et al., 2023c)

to avoid adversarial samples mirroring popular samples, as indoor scenes often contain similar objects.

These sampling strategies are designed to challenge the model's robustness and assess its susceptibility to different levels of object hallucination.

**Metrics.** To evaluate the model's performance on the 3D-POPE benchmark, we use key metrics including *Precision*, *Recall*, *F1 Score*, *Accuracy*, and *Yes (%)*. *Precision* and *Recall* assess the model's ability to correctly affirm the presence of objects and identify the absence of objects, respectively. *Precision* is particularly important as it indicates the proportion of non-existing objects generated by the 3D-LLMs. The *F1 Score*, combining Precision and Recall, offers a balanced view of performance and serves as the primary evaluation metric. *Accuracy* measures the proportion of correctly answered questions, encompassing both "Yes" and "No" responses. Additionally, the *Yes (%)* metric reports the ratio of incorrect "Yes" responses to understand the model's tendencies regarding object hallucination.

**Leaderboard.** We establish a public leaderboard for the 3D-POPE benchmark, allowing researchers to submit their 3D-LLM results and compare their performance against other state-of-the-art models. The leaderboard reports the evaluation metrics for each model under the three sampling strategies, providing a transparent and standardized way to assess the hallucination performance of 3D-LLMs.

## 5 EXPERIMENTS

In this section, we present our experimental setup, including the baselines, datasets, and implementation details. We then report the results of our approach, denoted as **3D-GRAND** on the ScanRefer (Chen et al., 2020) and the 3D-POPE benchmark, demonstrating the effectiveness in improving grounding and reducing hallucination. Finally, we conduct an ablation study to analyze the impact of different components of our model and training strategy.

### 5.1 EXPERIMENTAL SETUP

**Model.** Our proposed model is based on Llama-2 (Touvron et al., 2023). The input is object-centric context, including a scene graph with each object's category, centroid (x, y, z), and extent (width, height, depth), along with the text instruction and user query. During training, we utilized ground-truth centroids and extents. For inference, we used bounding boxes predicted by Mask3D (Schult et al., 2023a). Examples of input/output and details of the model can be found in the supplementary material.

**Baselines.** We compare our 3D-GRAND against the following baselines: 3D-LLM (Hong et al., 2023b), LEO (Huang et al., 2024), and 3D-Vista (Zhu et al., 2023c). Each model, along with the specific checkpoint used to obtain the results, is documented in the appendix.

**Datasets.** We evaluate our model 3D-GRAND on two datasets: 3D-POPE and ScanRefer. 3D-POPE is our newly introduced benchmark dataset for evaluating object hallucination in 3D-LLMs, as described in Section 4. For ScanRefer, We utilized the validation split which contains 9,508 natural language descriptions of 2,068 objects in 141 ScanNet (Dai et al., 2017) scenes.

**Metrics.** For the ScanRefer benchmark, we use the official evaluation metrics, including Accuracy@0.25IoU and Accuracy@0.5IoU. For the 3D-POPE benchmark, we report accuracy, precision, recall, F1 score, and "Yes" rate under the three sampling strategies described in Section 4.

**Implementation Details.** The 3D-GRAND model is LoRA-finetuned (Hu et al., 2022) based off Llama-2. We use DeepSpeed ZeRO-2 (Rasley et al., 2020) and FlashAttention (Dao, 2024) to save GPU memory and speed up training. The model is trained in BF16 precision on 12 NVIDIA A40 GPUs with a combined batch size of 96 and a learning rate of 2e-4. We use the AdamW (Loshchilov & Hutter, 2019) optimizer with a weight decay of 0.01 and a cosine learning rate scheduler. We train the mode for 10k steps, which takes approximately 48 hours.

### 5.2 RESULTS ON 3D-POPE

We first evaluate these approaches on 3D-POPE and report results on Table 5. Results show that 3D-LLM (Hong et al., 2023b) almost always produces *yes* responses to any question. 3D-VisTA (Zhu et al., 2023c) performs similarly to the random baseline. LEO (Huang et al., 2024) tends to answer *yes* frequently, but its precision indicates a similar object hallucination rate to the random baseline. In our evaluation, 3D-GRAND achieved exceptional performance, with 93.34% precision and 89.12%

| Model | Generative 3D-LLM? | Never seen ScanNet? | Acc@0.25 | Acc@0.5 |
|---|---|---|---|---|
| **Non-LLM based** | | | | |
| ScanRefer | ✗ | ✗ | 37.3 | 24.3 |
| MVT | ✗ | ✗ | 40.8 | 33.3 |
| 3DVG-Trans | ✗ | ✗ | 45.9 | 34.5 |
| ViL3DRel | ✗ | ✗ | 47.9 | 37.7 |
| M3DRef-CLIP | ✗ | ✗ | **51.9** | **44.7** |
| **Non-Generative 3D-LLMs** | | | | |
| 3D-VisTA (zero-shot) | ✗ | ✓ | 33.2 | 29.6 |
| SceneVerse (zero-shot) | ✗ | ✓ | **35.2** | **31.1** |
| **Generative 3D-LLMs** | | | | |
| 3D-LLM | ✓ | ✗ | 30.3 | - |
| LLM-Grounder | ✓ | ✓ | 17.1 | 5.3 |
| 3D-GRAND (Ours) | ✓ | ✓ | **38.0** | **27.4** |

Table 6: ScanRefer Results for evaluating visual grounding capability of 3D-LLMs. 3D-GRAND achieves the best zero-shot performance among 3D-LLMs, providing signals for sim-to-real transfer.

| Method | Det. | Unique | | Multiple | | Overall | |
|---|---|---|---|---|---|---|---|
| | | Acc@0.25 | Acc@0.5 | Acc@0.25 | Acc@0.5 | Acc@0.25 | Acc@0.5 |
| Best IoU (upper bound) | Mask3D (Top100) | 93.7 | 66.8 | 91.6 | 70.7 | 92.4 | 69.2 |
| Best IoU (upper bound) | Mask3D (Top40) | 81.2 | 58.7 | 80.7 | 62.4 | 80.9 | 61.0 |
| Non-grounded Model | Mask3D (Top40) | 51.8 | 33.1 | 21.3 | 17.9 | 34.2 | 24.3 |
| Grounded Model (ground later) | Mask3D (Top40) | 50.4 | 32.4 | 26.0 | 20.5 | 36.3 | 25.5 |
| Grounded Model (ground first) | Mask3D (Top40) | **54.4** | **36.4** | **26.0** | **20.8** | **38.0** | **27.4** |
| Best IoU (upper bound) | GT | 100.0 | 100.0 | 100.0 | 100.0 | 100.0 | 100.0 |
| Non-grounded Model | GT | 90.8 | 90.8 | 26.0 | 26.0 | 53.4 | 53.4 |
| Grounded Model | GT | **91.0** | **91.0** | **32.1** | **32.1** | **57.0** | **57.0** |

Table 7: Ablation Study on Grounding Accuracy (%) on ScanRefer: Training with densely-grounded data significantly improves grounding accuracy, particularly when multiple distractor objects of the same category are present in the room.

accuracy when tested with random sampling. However, our model struggles with the more difficult splits, Popular and Adversarial, which demonstrates the effectiveness and rigorousness of 3D-POPE as a benchmark. Moreover, we emphasize that our model has never encountered ScanNet during training. More analysis on 3D hallucination can be found in the supplementary material.

## 5.3 RESULTS ON SCANREFER

We report results on ScanRefer in Table 6. There are a few important observations on this result:
- Our 3D-LLM trained with 3D-GRAND data achieved the best Acc@0.25 among all models. Notably, our model surpasses the previous best-performing model, 3D-LLM, by 7.7% on accuracy@0.25IoU. We emphasize that our model, unlike 3D-LLM, has never seen ScanNet scenes during its training (zero-shot) and is only trained on synthetic 3D scenes instead of real scans. Therefore, these results provide a promising early signal that sim-to-real transfer can be achieved via our densely-grounded large-scale dataset.
- Our generative 3D-LLM model (one that a user can chat with) performs better or on par compared to non-generative 3D-LLMs such as 3D-VisTA and SceneVerse. In the past, generative 3D-LLMs are usually significantly outperformed by non-generative 3D-LLMs, as the latter usually sacrificed the ability to chat in exchange for incorporating specialized model designs, such as producing scores for each object candidate. These designs are closer to traditional non-LLM-based specialized models. But here, we observe that the gap between the two modeling choices is closing with the help of large-scale densely-grounded data like 3D-GRAND.
- It is worth noting that our model is just a naive text-based model (Sec. 5.1) to demonstrate the effectiveness of the dataset - in our model, little visual information is conveyed between the mask proposal to the LLM, contrast to some of the other more sophisticated models where 3D object embeddings are used to better represent visual information. This means 3D-GRAND as a dataset has more potential to be unlocked in the future.

## 5.4 ABLATION STUDY

To better understand the impact of different components of our 3D-LLM, we conduct an ablation study on the ScanRefer and 3D-POPE benchmarks.

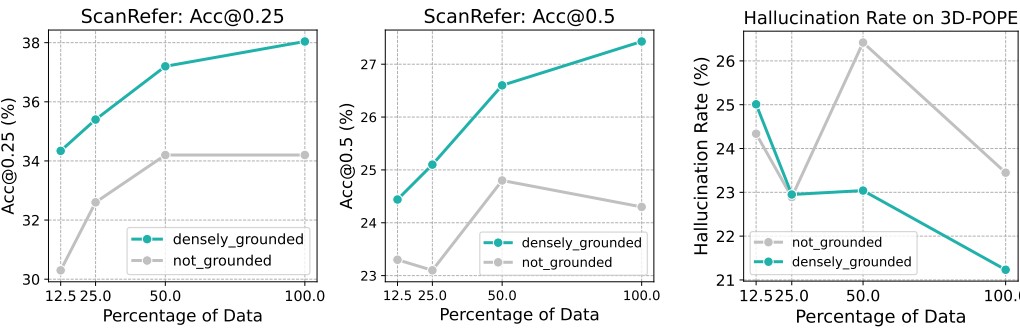

(a) Grounding capability. Higher is better.      (b) Hallucination rate. Lower is better.

Figure 4: **Data scaling analysis** on zero-shot, sim-to-real grounding capability, and hallucination. Grounding performance consistently improves as data scales up. Model trained with densely-grounded data exhibits better grounding capability compared to that trained without. Additionally, model hallucinates less when exposed to more data from 3D-GRAND. Here, the Hallucination Rate is calculated as $(1 - \text{Precision})$ on 3D-POPE.

**Grounding tokens.** We show the results of our model with different types of grounding methods in Table 7. We also show results on 3D-POPE in Table 8. In general, the model has a worse grounding performance and more hallucinations without grounding tokens. "Ground First" and "ground later" refer to whether the dense grounding (grounding every single object mentioned) of the object reference query happens before or after the model outputs the final answer for the refer expression. The former effectively constitutes a chain-of-thought reasoning process (Wei et al., 2022), which is likely why the performance increases compared to the latter. See Appendix for details.

| 3D-POPE | Model | Precision |
|---|---|---|
| *Random* | 3D-GRAND | 93.34 |
| | w/o grounding tokens | (-1.38) |
| *Popular* | 3D-GRAND | 73.05 |
| | w/o grounding tokens | (-2.68) |
| *Adversarial* | 3D-GRAND | 69.86 |
| | w/o grounding tokens | (-2.38) |

Table 8: Ablation on 3D-POPE. Without the grounding tokens, 3D-GRAND hallucinates more.

**Mask3D proposals.** Finally, we show the upper bound of our approach in Table 7. Our results are based on Mask3D proposals. Due to the context length of LLM, we only use top-40 proposals.

### 5.5 DATA SCALING AND SIM-TO-REAL TRANSFER

The results are presented in Figure 4. Our model is trained on synthetic 3D scenes from 3D-FRONT and Structured3D (Zheng et al., 2020; Fu et al., 2021), and evaluated on real-world 3D scans from ScanNet (Dai et al., 2017). The grounding performance consistently improves, and the hallucination rate drops as the densely-grounded data scales up. Notably, our model trained on densely grounded data scales better than the same model trained without such data. These findings pave the way for a future where we can scale 3D-text understanding using synthetic scenes obtained from simulation, which is much cheaper and more accessible to obtain.

## 6 CONCLUSION

In this paper, we introduced 3D-GRAND, a large-scale, densely-grounded 3D-text dataset designed for grounded 3D instruction tuning, and 3D-POPE, a comprehensive benchmark for evaluating object hallucination in 3D-LLMs. Through extensive experiments, we demonstrated the effectiveness of our dataset on 3D-LLMs in improving grounding and reducing hallucination, achieving state-of-the-art performance on the ScanRefer and 3D-POPE benchmarks. Our ablation study and qualitative analysis highlighted the importance of densely-grounded instruction tuning, the data scaling law, and effective sim-to-real transfer in developing high-performing 3D-LLMs. We hope our contributions and findings can spark further research and innovation in this field, ultimately leading to the development of more advanced and capable 3D-LLMs for a wide range of applications.

### ETHICS STATEMENT

All 3D indoor scene data used to produce 3D-GRAND are publicly available data that do not contain any personal information. We have manually and programmatically examined the produced text data and made sure they do not contain any profanity or harmful languages.

REPRODUCIBILITY STATEMENT

All data of 3D-GRAND and 3D-POPE will be made free and publicly available with a permissive license for non-commercial usage. We have also set up infrastructure (e.g., via HuggingFace Datasets) to host the data to ensure long-term accessibility. All code used to produce the results in the paper is available in the supplementary material. The code and model weights will also be open-sourced.

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

## A   IMPLEMENTATION OF OUR MODEL

### A.1   MODEL INPUT AND OUTPUT DEMONSTRATION

In Figure 5, we show an example of 3D-GRAND model's input and output on the Grounded Object Reference task. Note how in the "Response", we train the model to generate a ⟨detailed_grounding⟩ pair of tags to densely ground every single object mentioned in the refer expression after generating the grounding answer in ⟨refer_expression_grounding⟩. The "ground first" "ground later" in Table 7 means whether the ⟨detailed_grounding⟩ tags happen before or after the ⟨refer_expression_grounding⟩ tags. Figure 5 is an example of "ground later", and Figure 6 shows an example of "ground first".

```
Enter the scene ID or type 'exit' to quit:: 9a6efb10-bdba-46e5-85e8-75993fd1235a@OtherRoom-116820
Enter your message for scene '9a6efb10-bdba-46e5-85e8-75993fd1235a@OtherRoom-116820' or type 'change' to switch scenes or 'exit' to
quit:: Ground these sentences: <refer_expression>This footstool has a striking blue and white striped pattern. With its round cushioned t
op and sleek metal legs, the blue and white striped footstool offers modern comfort. It is positioned below the pendant lamp and is place
d in front of the double bed.</refer_expression>
───────────────────────────── Prompt ─────────────────────────────
Model prompt: [INST] <<SYS>>
You are a helpful language and vision assistant that helps human reason about a 3D scene. You will be provided with a 3D scene
context that contains a set of objects. You are able to understand the 3D scene, and answer the user's questions truthfully with
respect to the 3D scene. In the 3D scene context, each object is represented by a unique identifier, such as <obj_0>, <obj_1>, etc.
In all of your responses, you should explicitly ground each object noun phrase to the corresponding object identifier in the 3D
scene.For example, when the user asks 'Describe this room.', you should respond with this format: 'This is a living room with a
<p>brown sofa</p>[<obj_2>] and <p>three chairs</p>[<obj_0>, <obj_1>, <obj_3>].'Note that your answer should always enclose object
noun phrases with <p> and </p> tags, followed by the corresponding object identifiers in square brackets.
<</SYS>>

%%%% Object-centric context: <obj_0>: {'category': 'pendant lamp', 'centroid': '[-3.97, 2.41, -0.25]', 'extent': '[0.81, 0.80,
0.81]'}; <obj_1>: {'category': 'nightstand', 'centroid': '[-2.43, 0.31, -1.70]', 'extent': '[0.46, 0.60, 0.53]'}; <obj_2>:
{'category': 'double bed', 'centroid': '[-3.29, 0.51, -0.70]', 'extent': '[2.19, 1.00, 1.58]'}; <obj_3>: {'category': 'desk',
'centroid': '[-5.41, 0.50, -0.02]', 'extent': '[0.66, 0.98, 1.12]'}; <obj_4>: {'category': 'footstool/sofastool/bed end stool/stool',
'centroid': '[-4.91, 0.23, -0.44]', 'extent': '[0.49, 0.45, 0.49]'}; <obj_5>: {'category': 'dining table', 'centroid': '[-4.98, 0.20,
0.25]', 'extent': '[0.48, 0.38, 0.73]'}; <obj_6>: {'category': 'footstool/sofastool/bed end stool/stool', 'centroid': '[-2.96, 0.27,
1.15]', 'extent': '[0.62, 0.52, 0.55]'};
Ground these sentences: <refer_expression>This footstool has a striking blue and white striped pattern. With its round cushioned top
and sleek metal legs, the blue and white striped footstool offers modern comfort. It is positioned below the pendant lamp and is
placed in front of the double bed.</refer_expression> [/INST]
──────────────────────────── Response ────────────────────────────
Model response: <refer_expression_grounding>These sentences refer to <p>the footstool</p>[<obj_4>].</refer_expression_grounding>
<detailed_grounding>This <p>footstool</p>[<obj_4>] has a striking blue and white striped pattern. With its round cushioned top and
sleek metal legs, the <p>blue and white striped footstool</p>[<obj_4>] offers modern comfort. It is positioned below the <p>pendant
lamp</p>[<obj_0>] and is placed in front of the <p>double bed</p>[<obj_2>].</detailed_grounding>
```

Figure 5: 3D-GRAND model input and output on Grounded Object Reference task.

### A.2   TRAINING DATA

There are two flavors of models that we fine-tuned: one grounded object reference model, and one grounded QA model. The grounded object reference model was trained using the grounded object reference data on 3D-FRONT train split, which consist of 234,791 3D-text pairs, each of which are densely grounded. This model was used to generate the ScanRefer results presented in Table 6, 7, and Figure 4 The grounded QA model was trained using a subset of 200k grounded QA: object existence data from the 3D-FRONT train split. The reason that we select a subset of 200k QAs is simply because the entire grounded QA dataset is too large and we do not have enough resource to train on all data. However, as shown in Table 5 and Figure 4, we find even such a subset is very effective in reducing hallucinations in 3D-LLMs.

We provide official data splits of train, val and test (90%, 5%, 5%) in our dataset release. The val and test proportion might seem small, but given our dataset's million-scale, they should be sufficiently large for any development and evaluation purposes.

### A.3   TRAINING DETAILS

The two flavors of model mentioned above are LoRA-finetuned (Hu et al., 2022) based off Llama-2. We use DeepSpeed ZeRO-2 (Rasley et al., 2020) and FlashAttention (Dao, 2024) to save GPU memory and speed up training. The model is trained in BF16 precision on 12 NVIDIA A40 GPUs with a combined batch size of 96 and a learning rate of 2e-4. We use the AdamW (Loshchilov & Hutter, 2019) optimizer with a weight decay of 0.01 and a cosine learning rate scheduler. We train the mode for 10k steps, which takes approximately 48 hours.

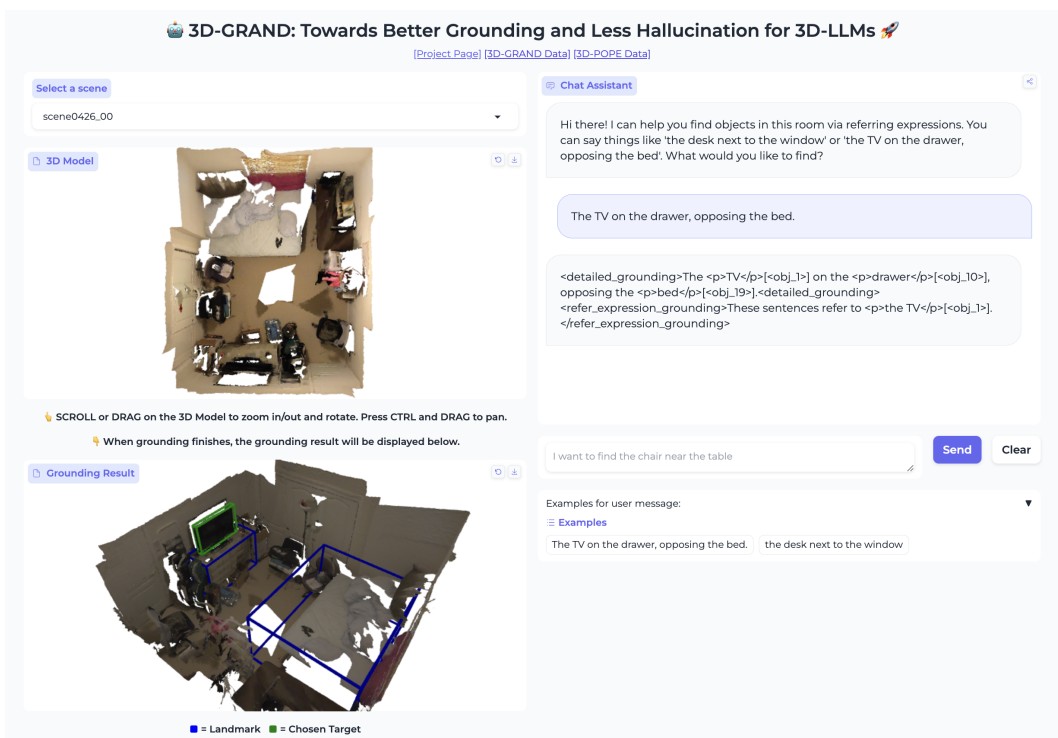

Figure 6: Demo of interactive chat interface with the 3D-GRAND model.

## B ADDITIONAL 3D-GRAND DATA COLLECTION

### B.1 POINT CLOUD GENERATION PIPELINE FOR 3D-FRONT

Here, we present an expanded version of Section 3, focusing on the methodologies employed in the collection and cleaning of 3D scenes, specifically detailing our process for deriving 3D point clouds from existing datasets.

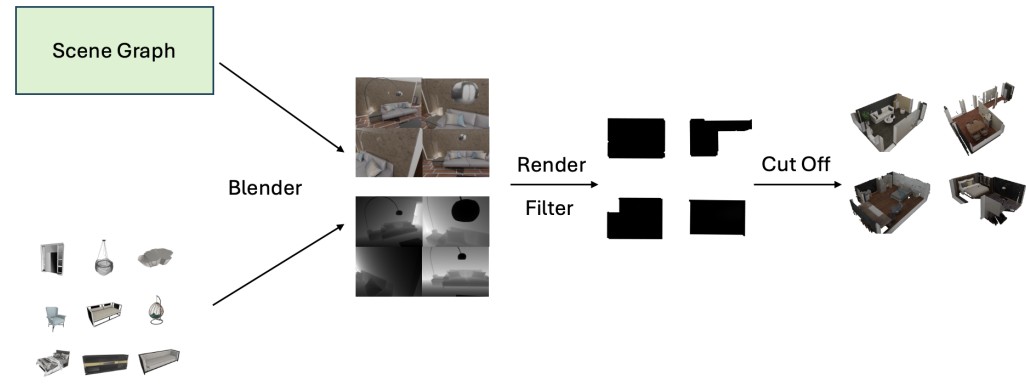

Figure 7: Point Cloud Generation for 3D-Front.

In our workflow with 3D-FRONT, layouts and meshes are initially processed in Blender to produce multi-view images. These images are subsequently used to construct comprehensive point clouds for entire houses. Both point clouds and per-room meshes are utilized to generate scene-level point clouds. We avoid direct use of room meshes because they lack color information in ceilings, walls, and floors, necessitating the final output to be a point cloud.

For Structure3D, while per-scene multi-view images facilitate direct rendering of per-scene point clouds, we frequently encounter issues where parts of adjacent scenes are inadvertently reconstructed due to window transparency. To address this, we employ the layout of each scene to trim extraneous points, thus enhancing the precision of the resulting point clouds.

## C ADDITIONAL 3D POPE RESULTS

### C.1 3D POPE RESULTS ON NYU40

Table 9 presents evaluation results for 3D POPE using the NYU40 class set. NYU40 includes a subset of the classes from ScanNet200 featured in the main results table. The NYU40 class set consolidates many fine-grained classes into an "other" category, potentially reducing the challenge of negative sampling in the *Popular* and *Adversarial* settings compared to the ScanNet200 scenario.

| Dataset | 3D-POPE | Model | Accuracy | Precision | Recall | F1 Score | Yes (%) |
|---|---|---|---|---|---|---|---|
| ScanNet Val (NYU40) | *Random* | 3D-LLM | 50.00 | 50.00 | 100.00 | 66.67 | 100.00 |
| | | 3D-VisTA | 50.12 | 50.08 | 77.13 | 60.73 | 77.01 |
| | | LEO | 54.03 | 52.70 | 78.52 | 63.07 | 74.50 |
| | | Ours zero-shot (No Grounding) | 86.45 | 87.26 | 85.36 | 86.30 | 48.91 |
| | | Ours zero-shot (Grounding) | 85.68 | 88.22 | 82.34 | 85.18 | 46.67 |
| | *Popular* | 3D-LLM | 50.00 | 50.00 | 100.00 | 66.67 | 100.00 |
| | | 3D-VisTA | 50.27 | 50.23 | 77.13 | 60.84 | 76.91 |
| | | LEO | 48.86 | 49.28 | 77.44 | 60.23 | 78.58 |
| | | Ours zero-shot (No Grounding) | 80.85 | 78.30 | 85.35 | 81.68 | 54.50 |
| | | Ours zero-shot (Grounding) | 81.69 | 81.32 | 82.28 | 81.80 | 50.59 |
| | *Adversarial* | 3D-LLM | 50.00 | 50.00 | 100.00 | 66.67 | 100.00 |
| | | 3D-VisTA | 50.44 | 50.48 | 77.14 | 61.03 | 76.86 |
| | | LEO | 49.77 | 49.85 | 77.67 | 60.73 | 77.91 |
| | | Ours zero-shot (No Grounding) | 81.47 | 78.98 | 85.78 | 82.24 | 54.31 |
| | | Ours zero-shot (Grounding) | 82.10 | 81.72 | 82.72 | 82.22 | 50.61 |

Table 9: Results of 3D-LLMs under three evaluation settings of 3D-POPE on the validation set of ScanNet using NYU40 class set. Yes denotes the proportion of answering "Yes" to the given question. The best results in each block are denoted in bold.

## D HUMAN VALIDATION

Because our dataset generation process involves GPT-4V, there is a potential for hallucinations. We identify three types of possible hallucinations that could impact our dataset: the text might inaccurately describe an object's property, such as color or size (termed incorrect object attribute); it might incorrectly depict the spatial relationship between two objects (termed incorrect spatial relation); or it might describe an object that does not exist in the referenced scene at all (termed incorrect object existence). Additionally, inaccuracies in our dataset may also arise from incorrectly grounding the wrong object.

To validate our dataset against these potential failures, we plan to verify a subset of our data through crowdsourcing to ascertain the frequency of these failure cases.

### D.1 CROWD-SOURCING

We crowd-source the validation of annotations using Hive, a platform commonly used for sourcing annotations for computer vision tasks. The platform can be accessed at https://thehive.ai/.

We conceptualize our dataset validation as a data annotation problem, employing scene-text pairs as the data unit. Annotators are instructed to label these pairs as "True" or "False" to indicate the presence or absence of hallucinations or inaccuracies. Additionally, a "Cannot Decide" option is provided to accommodate cases where the scene view is unclear.

### D.1.1 TASK GENERATION

Hive only supports presenting static images to annotators, so we generate annotation tasks by composing snapshots of a scene with corresponding text annotations. For each task, we take snapshots

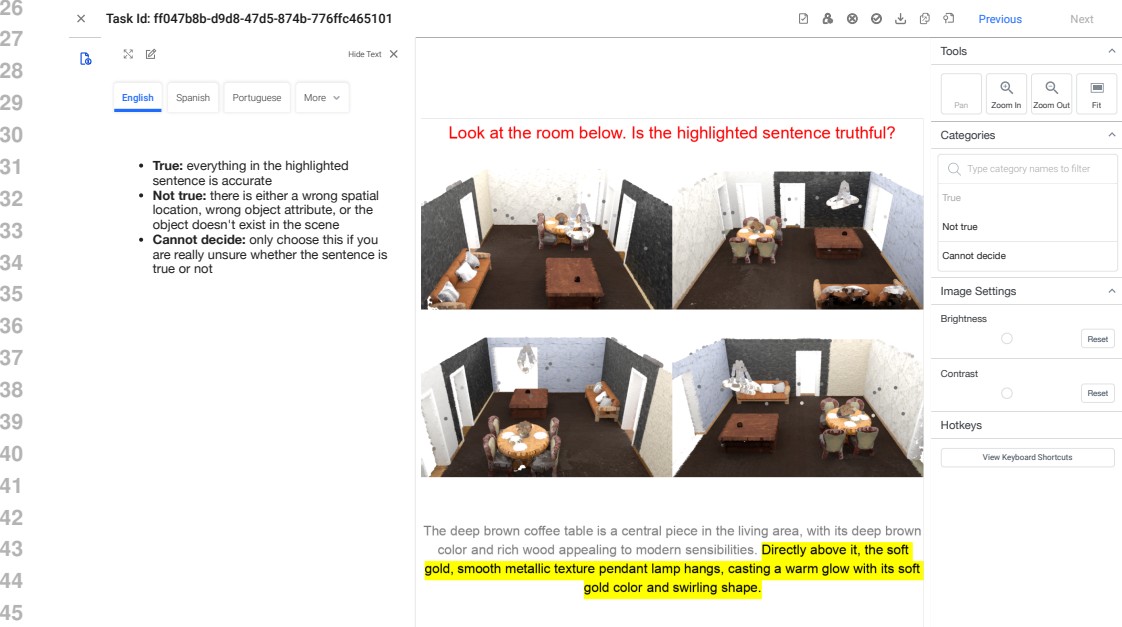

Figure 8: Example of a dataset validation task presented to crowd-sourcing annotators: It displays a scene from four different angles alongside the sentence to be validated, which is highlighted. Annotators have the options to select "True," "Not True," or "Cannot Decide." On the left side of the screen, the instructions are repeated for annotators' reference. In this example, "True" is selected.

from four different angles and pair them with a corresponding annotation. To maintain simplicity and conciseness, we require validation of just one sentence per task, providing some surrounding context and highlighting the target sentence. For grounding validation, the grounded object is outlined in the scene with a bounding box, and the referring phrase in the sentence is emphasized. An example of such a task, along with the annotation interface, is depicted in Figure 8. Figure 9 displays two text validation tasks and two grounding validation tasks that were presented to annotators.

### D.1.2 CROWD-SOURCING VALIDITY

Validating a dataset necessitates a high level of attention from annotators. We curate sets of instructions, qualifying tasks, and honeypot tasks to ensure that the annotations obtained from crowdsourcing are reliable. The crowdsourcing process is illustrated in Figure 10.

Before presenting any tasks to the workers, we present them with a set of instruction tasks that show an example annotation, the correct response (as determined by us), and the reason why that response is correct. They are paired with an incorrect example and an explanation of why it is incorrect in order to ensure unbiased annotations. Examples of qualifying instructions are shown in 11. These instructions are intentionally brief, as we found through trial-and-error that longer, paragraph-based instructions were largely ignored by annotators.

Qualifying tasks are presented to the annotators before they are shown any real tasks in order to train them to complete the real task with a high accuracy. Annotators are both shown the correct answer and a reasoning as to why it is correct for every qualifier. We set the minimum qualifier accuracy to 0.75 to ensure that annotators must achieve a minimum competency before annotating real tasks. Every dataset is given between 12 and 30 specially crafted qualifying tasks that demonstrate the possible inaccuracies that could appear in the data. These qualifiers are divided equally between true and false examples so as not to bias workers towards any one answer.

Honeypot tasks are randomly mixed in with real tasks in order to ensure that annotators are maintaining a high quality of annotations throughout the entire job. Because we annotate the honeypot tasks before showing them to annotators, we are able to evaluate any given worker's accuracy on honeypot tasks. We set the minimum honeypot accuracy to 0.89 to ensure that annotators are maintaining

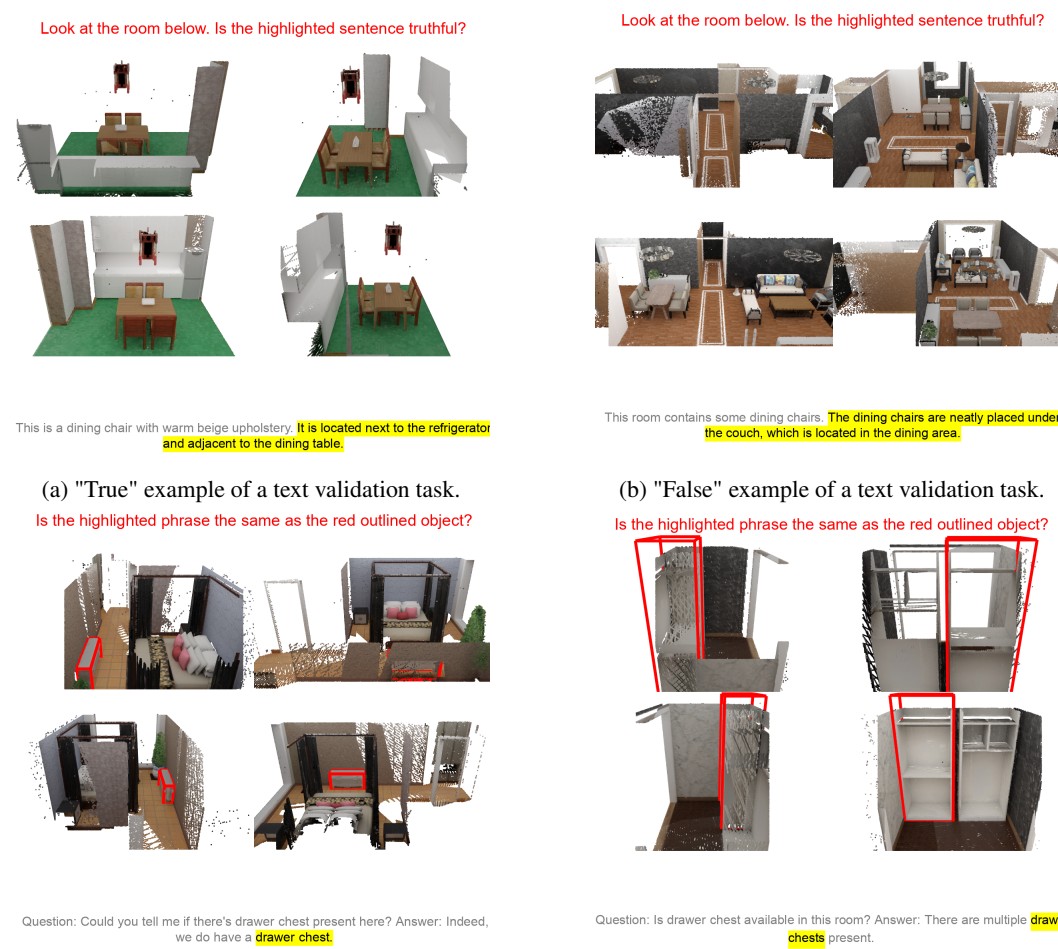

(a) "True" example of a text validation task.

(b) "False" example of a text validation task.

(c) "True" example of a grounding validation task.

(d) "False example of a grounding validation task.

Figure 9: Examples of tasks presented to annotators for validating both text accuracy and grounding accuracy. The instruction is displayed at the top of the task, while the center showcases four different views of the scene to ensure comprehensive coverage of all relevant areas. At the bottom, the annotation is presented with the pertinent section highlighted.

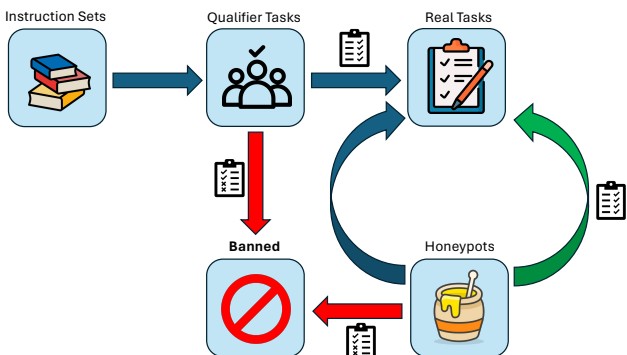

Figure 10: Illustration of the crowd-sourcing process. Annotators are first shown instruction sets that describe both how the task should be completed and the possible inaccuracies that could appear in the data. They are then presented with qualifier tasks, and annotators who do not get a high enough accuracy on these tasks are banned from annotating our dataset. Annotators who pass the qualifier are able to annotate real tasks, but are randomly presented with honeypots that are indistinguishable from real tasks. Annotators who do not get a high enough accuracy on honeypots are also banned from our dataset.

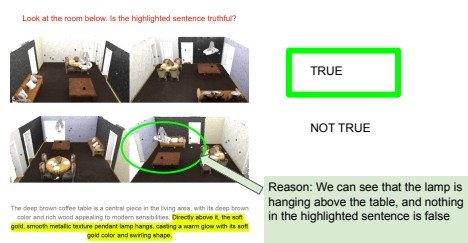

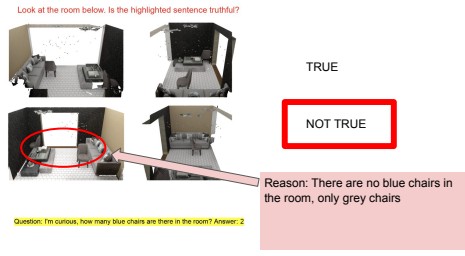

(a) "True" example instruction for the text valida-tion task.

(b) "False" example instruction for the text valida-tion task.

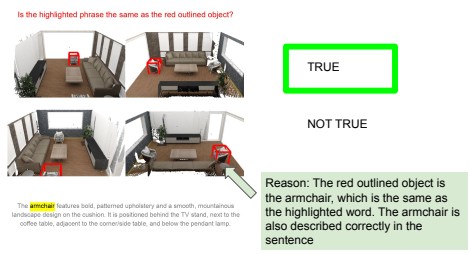

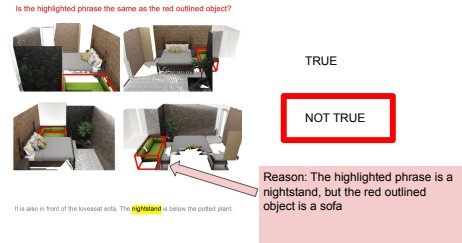

(c) "True" example instruction for the grounding validation task.

(d) "False" example instruction for the grounding validation task.

Figure 11: Examples of instructions presented to annotators before they are shown any actual tasks for annotation. For every possible kind of hallucination (incorrect object attribute, spatial relation, or object existence), an illustrative positive and negative example are presented in order to instruct the annotator to look for all possible failure cases.

correct annotations. Workers that do not maintain this accuracy are banned from annotating our tasks. This is higher than the required accuracy for qualifiers because we expect annotators to already be well trained in our annotation tasks from the instructions and qualifiers. Every data type is given between 18 and 35 honeypot tasks. The honeypots are also approximately divided equally between true and false examples so that workers who consistently select a single answer without paying attention to the task (e.g., someone who always selects "True") will be banned.

To further ensure high-quality annotations, we send each question to 3 different annotators and only accept an annotation if at least 2 out of the 3 annotators agree with each other on the truthfulness of an item. If agreement is not reached, the task is returned as inconclusive.

## D.2 RESULTS

We perform validation on 10,200 room-annotation pairs. From each of the three data types, 1,700 pairs are sampled for validation of both text truthfulness and grounding accuracy. A subset of 800 rooms is uniformly chosen, with 400 designated for text truthfulness and another 400 for grounding accuracy. The text data is uniformly sampled from these rooms. We report accuracies for both text truthfulness and grounding accuracy in Table 10.

We report comprehensive statistics from the annotation process in Table 11. We observe a very low qualifier pass rate ranging from 11 - 20 % across the different tasks in our data, suggesting that our qualifiers were effective in allowing only the most attentive annotators qualify to annotate real tasks. In addition, none of these annotators were banned due to honeypots. This increases our confidence that our qualification process is effective in training annotators and filtering out those who were not attentive. We also observe that workers spend roughly the same time on real tasks and honeypot tasks, suggesting that the honeypots are indistinguishable from real tasks for the annotators. This further supports the validity of our annotations.

Table 10: Text Truthfulness and Grounding Accuracy from crowdsourcing. Accuracy is computed by dividing the number of "True" responses by the total number of tasks (1700).

| Method | Text Truthfulness | Grounding Accuracy |
|---|---|---|
| Grounded Scene Description | 0.877 | 0.944 |
| Grounded QA | 0.852 | 0.956 |
| Grounded Object Reference | 0.863 | 0.918 |

Table 11: Comprehensive annotation metrics. Includes qualifier pass rate, honeypot count, honeypot ban rate, percent of tasks marked inconclusive (where workers could not come to an agreement on the label), and the average time that workers spend on both real tasks and honeypot tasks. Each dataset was evaluated on 1700 annotations. At least 2 workers must agree on the label for an annotation to be considered valid.

| Category | Type | % Qualifier Pass Rate | # Honeypots | % Honeypot Ban Rate | % of Inconclusive Tasks | Avg. Real Task Speed (s) | Avg. Honeypot Speed (s) |
|---|---|---|---|---|---|---|---|
| | | (Pass) | (Total) | (Ban) | (Tasks) | (Real) | (Honeypot) |
| Text Accuracy | Scene Description | 17 | 35 | 0 | 2.03 | 17.91 | 17.08 |
| | QA | 19 | 18 | 0 | 1.35 | 16.22 | 16.64 |
| | Object Reference | 10 | 38 | 0 | 0.82 | 19.88 | 17.57 |
| Grounding Accuracy | Scene Description | 20 | 18 | 0 | 1.66 | 9.69 | 12.84 |
| | QA | 16 | 18 | 0 | 1.11 | 6.65 | 11.14 |
| | Object Reference | 20 | 18 | 0 | 1.88 | 9.69 | 12.84 |