# OpenReview forum: "3D-GRAND: A Million-Scale Dataset for 3D-LLMs with Better Grounding and Less Hallucination"
_ICLR.cc/2025/Conference — ICLR 2025 Conference Withdrawn Submission_

### Official Review · Reviewer_PFXv · 2024-10-27

**Soundness:** 2
**Presentation:** 2
**Contribution:** 2
**Rating:** 3
**Confidence:** 4

**Summary:**

This paper introduces 3D-GRAND, a large-scale dataset comprising 6.2 million densely scene-grounded data instances. The dataset is built upon existing synthetic 3D assets, including 3D-Front and Structured3D. The authors have developed a data generation pipeline based on scene graphs, which have been proposed in previous works [1][2]. Additionally, the paper designs a benchmark called 3D-POPE to facilitate object hallucination. A benchmark 3D-POPE is designed based on experiences from 2D Vision-Language Models (VLMs) [3]. The experiment results reveal the limitation of existing 3D-VLMs in object hallucination. The paper proposes a large language model (LLM) named 3D-GRAND and evaluates its performance on the dataset compared to other 3D VLMs. However, no existing 3D VLMs have been trained on the large-scale 3D-GRAND dataset, leaving its impact on these models unknown.

In summary, while the dataset presents a scene graph-based data generation pipeline, its overall technical contribution to dataset collection is weak. The paper also proposes a new benchmark to evaluate the object hallucination of 3D VLMs. However, no existing baseline models have been trained on the large-scale 3D-GRAND dataset, which leads to a lack of analyses for the dataset and models. Consequently, the experiment results do not provide useful insights for the community to improve model design.

[1] An Embodied Generalist Agent in 3D World
[2] SceneVerse: Scaling 3D Vision-Language Learning for Grounded Scene Understanding
[3] Evaluating Object Hallucination in Large Vision-Language Models

**Strengths:**

1. The scale of the language instances is significantly larger than that of previous works, such as SceneVerse and EmbodiedScan.
2. The authors conducted manual and programmatic reviews of the dataset samples and provided statistical results related to data quality. The details of the data generation process are clearly presented, making it easy to understand.
3. The evaluation results emphasize the problem of object hallucination in existing 3D-VLMs for 3D applications.

**Weaknesses:**

1. LEO [1] and SceneVerse [2] have proposed the scene graph-based data generation pipeline. The pipeline's technical contribution is very minor compared to previous work.
2. No existing 3D VLM is trained based on 3D-GRAND dataset. The author only mentions that they train 3D-GRAND models based on a specific subset of 3D-FRONT in the appendix. There should be 3D VLMs(existing models such as 3D-LLM, LEO, etc.) trained based on 3D-GRAND as evidence of the quality of the dataset. This results in poor integrity of the article and greatly reduces the contribution of the article.
3. Unfair comparison with the baseline models. The proposed baseline model 3D-GRAND takes the object locations and attributes as the input, which provides strong information on the whole 3D scene without any noise. While other models such as 3D-LLM, 3D-Vista, and LEO take the 3D point cloud or multi-view images as input.
4. The training set of the baseline model 3D-GRAND should be presented in section 3 instead of in the appendix.

[1] An Embodied Generalist Agent in 3D World
[2] SceneVerse: Scaling 3D Vision-Language Learning for Grounded Scene Understanding

**Questions:**

Relevant 3D vision-language datasets and benchmarks such as EmbodiedScan[1], SceneVerse[2], MMScan[3] and MSR3D[4] have trained baseline 3D VLMs on their large-scale dataset and provide a fair comparison. Why is no existing 3D VLM trained on 3D-GRAND? Are there any underlying limitations or challenges that have not been explicitly discussed in the paper? Additionally, does the performance improvement of the proposed 3D-GRAND model over other baselines arise primarily from the privileged inputs (e.g., object locations and attributes) or the pretraining on the 3D-GRAND dataset?

[1] EmbodiedScan: A Holistic Multi-Modal 3D Perception Suite Towards Embodied AI
[2] SceneVerse: Scaling 3D Vision-Language Learning for Grounded Scene Understanding
[3] MMScan: A Multi-Modal 3D Scene Dataset with Hierarchical Grounded Language Annotations
[4] Multi-modal Situated Reasoning in 3D Scenes

---

### Official Review · Reviewer_DquU · 2024-11-01

**Soundness:** 3
**Presentation:** 3
**Contribution:** 3
**Rating:** 6
**Confidence:** 5

**Summary:**

The paper presents 3D-GRAND, a large-scale dataset designed to advance grounded language understanding in 3D-based large language models (3D-LLMs). With over 40,000 household scenes and 6.2 million densely-grounded scene-language pairs, 3D-GRAND aims to improve grounding accuracy and reduce object hallucination—a common issue in language models interpreting 3D environments. Alongside 3D-GRAND, the authors introduce 3D-POPE, a benchmark specifically designed to evaluate hallucination in 3D-LLMs. The paper demonstrates the effectiveness of training 3D-LLMs with 3D-GRAND by showing that the dataset significantly enhances grounding capabilities and reduces hallucinations. Results also indicate promising sim-to-real transfer capabilities, suggesting that synthetic data training could translate well into real-world performance. Together, 3D-GRAND and 3D-POPE contribute essential resources to the embodied AI community and provide a foundation for further research in 3D-LLM grounded reasoning.

**Strengths:**

1) Innovative Large-Scale Dataset for 3D-Language Grounding:
3D-GRAND is a pioneering contribution in its scale and grounding precision, filling a critical gap in the resources available for 3D-LLMs. Its dense grounding annotations at the noun-phrase level improve object-level clarity and enable the model to better understand spatial relations and attributes within complex scenes. The dataset’s vast scale allows for diverse applications, from spatial QA to scene description, establishing a robust foundation for training 3D-LLMs.

2) Comprehensive Hallucination Benchmark with 3D-POPE:
The introduction of 3D-POPE as a dedicated benchmark for 3D object hallucination is a valuable addition. Its well-defined metrics, including precision, recall, and F1 scores, allow for standardized, transparent evaluation of hallucination tendencies across different 3D-LLMs. The benchmark’s structured evaluation through varied sampling (random, popular, and adversarial) makes it rigorous, challenging models to correctly identify the presence or absence of objects even under difficult conditions.

3) Effective Use of Synthetic Data for Cost and Scalability:
By leveraging synthetic 3D scenes and automated annotation, 3D-GRAND achieves substantial cost and time savings over traditional human-annotated datasets. The paper highlights the efficiency of using large language models (GPT-4) for annotation, which makes scaling feasible and reduces labor-intensive work while maintaining a low annotation error rate. This approach supports large-scale experimentation and promotes accessibility for researchers working with limited resources.

4) Promising Signals for Sim-to-Real Transfer: The paper provides encouraging evidence that models trained on synthetic 3D-GRAND data perform well on real-world 3D scans, indicating potential for sim-to-real transfer. This transfer capability is crucial for embodied AI applications, such as robotics, where real-world adaptability is often required. By validating the dataset’s utility for real-world tasks, the authors strengthen the practical relevance of their work.

5) Extensive Experimental Analysis and Ablations: The authors conduct comprehensive experiments across multiple datasets, including 3D-POPE and ScanRefer, to demonstrate the benefits of densely-grounded data. Ablation studies further highlight the impact of various grounding techniques, such as the inclusion of grounding tokens, on hallucination and grounding accuracy. This level of experimental rigor provides strong evidence for the dataset’s effectiveness and aids in understanding the benefits and limitations of the approach.

**Weaknesses:**

1) While 3D-GRAND demonstrates early signs of sim-to-real transfer, it relies heavily on synthetic data from the 3D-FRONT and Structured3D datasets. There’s limited analysis on potential biases that might arise from this reliance, especially since synthetic scenes can differ significantly from real-world 3D environments in object layout and scene composition. This limitation could affect the model’s performance on real-world tasks where variability is higher.
2) Although the paper introduces a hallucination filter, the methodology behind it is not fully explained. For instance, it’s unclear how specific filtering criteria are determined, or how consistently the filter distinguishes valid object references from hallucinated ones. This could lead to inconsistencies in the data and, consequently, model performance. A more transparent breakdown of the filter’s effectiveness, or examples of common hallucination cases, would enhance confidence in the filtering approach.
3) Dense grounding annotations at the noun-phrase level are presented as beneficial, but the paper does not discuss potential downsides, such as increased model complexity or the risk of overfitting to these annotations. It’s also unclear whether dense grounding provides a significant advantage over sparse grounding for all task types or only specific ones. A comparison with sparse grounding on certain benchmarks could clarify this point.

**Questions:**

1) Given the reliance on synthetic scenes in 3D-GRAND, have you observed or tested any bias when generalizing to real-world 3D data, particularly regarding scene composition and object variability?
2) Could you elaborate on the criteria and process used by the hallucination filter to distinguish between correct and hallucinated object references?
3) Given the annotation error rate of 5.6–8.2%, could you discuss how these errors influence model predictions, especially in densely grounded tasks?

---

### Official Review · Reviewer_Wx4r · 2024-11-04

**Soundness:** 3
**Presentation:** 2
**Contribution:** 3
**Rating:** 6
**Confidence:** 5

**Summary:**

The paper introduces 3D-GRAND, a large-scale, densely grounded 3D-text dataset with 40,087 3D household scenes and 6.2 million text annotations for advancing 3D Large Language Models (LLMs) in understanding and grounding language in 3D environments. The dataset aims to improve the grounding capabilities of 3D-LLMs and reduce object hallucination, a common issue where models misidentify or invent objects in a scene.

**Strengths:**

- The paper introduces a large-scale, densely-grounded dataset, 3D-GRAND, comprising 40k 3D scenes and 6.2 million language annotations that support various grounded tasks such as object reference, scene description, and question-answering.
- 3D-POPE, a new benchmark, is introduced to systematically evaluate and quantify object hallucination in 3D-LLMs. By posing existence questions about objects within scenes, 3D-POPE enables measurement of hallucination rates and provides fair comparisons across models.
- The paper’s data collection pipeline leverages synthetic 3D scenes from the 3D-FRONT and Structured3D datasets, annotated using large language models like GPT-4 for dense grounding.
- Experimental results demonstrate that training with 3D-GRAND improves grounding performance and reduces hallucination, achieving superior results on benchmarks like ScanRefer and 3D-POPE. The authors conduct ablation studies and scaling analyses to highlight the impact of dense grounding data and explore sim-to-real transfer capabilities, establishing the dataset’s effectiveness in advancing 3D-LLMs.

**Weaknesses:**

- The benchmark primarily focuses on binary existence questions, lacking evaluation on more complex tasks such as spatial reasoning, attribute-based grounding, and relational understanding, which are essential for fully assessing a model’s hallucination in 3D comprehension.
- The proportion of data generated via templates versus LLM-generated annotations is unclear, and the paper does not provide specific examples of the templates used.
- How does the paper handle metric calculation in cases where objects in the model output are ungrounded? For example, if the model describes a “blue chair” that does not exist in the scene or fails to reference an existing object, how are grounding accuracy and hallucination rates calculated in such instances?
- In addition to grounding experiments, the paper could benefit from including language tokens prediction experiments, which would further demonstrate the dataset's effectiveness in training language understanding components of 3D-LLMs.
- The model design is relatively straightforward, relying on scene graphs to process 3D scenes and output JSON representations. This approach lacks advanced reasoning capabilities, which could limit the model’s ability to handle tasks requiring deeper understanding and inference in 3D environments.
- The figures in the paper need to be refined to improve their quality.

**Questions:**

See Weaknesses

---

### Note · Authors · 2024-11-12

I have read and agree with the venue's withdrawal policy on behalf of myself and my co-authors.